# Cell Differentiation and Proliferation in the Bone Marrow and Other Organs of 2D2 Mice during Spontaneous Development of EAE Leading to the Production of Abzymes

**DOI:** 10.3390/molecules27072195

**Published:** 2022-03-28

**Authors:** Kseniya S. Aulova, Andrey E. Urusov, Ludmila B. Toporkova, Sergey E. Sedykh, Juliya A. Shevchenko, Valeriy P. Tereshchenko, Sergei V. Sennikov, Irina A. Orlovskaya, Georgy A. Nevinsky

**Affiliations:** 1Institute of Chemical Biology and Fundamental Medicine, Siberian Branch of the Russian Academy of Sciences, 630090 Novosibirsk, Russia; amaya.rain.nsu@gmail.com (K.S.A.); urusow.andrew@yandex.ru (A.E.U.); sirozha@gmail.com (S.E.S.); 2Institute of Clinical Immunology, Siberian Branch of the Russian Academy of Sciences, 630090 Novosibirsk, Russia; toporkova12@mail.ru (L.B.T.); shevcen@ngs.ru (J.A.S.); tervp@ngs.ru (V.P.T.); sennikov_sv@mail.ru (S.V.S.); irorl@mail.ru (I.A.O.)

**Keywords:** 2D2, Th and C57BL/6 EAE mice, development of experimental autoimmune encephalomyelitis (EAE), hematopoietic stem cells differentiation, lymphocyte proliferation in different organs, catalytic antibodies

## Abstract

The exact cellular and molecular mechanisms of multiple sclerosis and other autoimmune diseases have not been established. Autoimmune pathologies are known to be associated with faults in the immune system and changes in the differentiation profiles of bone marrow stem cells. This study analyzed various characteristics of experimental autoimmune encephalomyelitis (EAE) in 2D2 mice. Differentiation profiles of six hematopoietic stem cells of bone marrow were found to significantly differ in 2D2 male and female mice during the spontaneous development of EAE. In addition, we found various properties of B and T cells, CD4+ and CD8+ lymphocytes in blood and several organs (bone marrow, spleen, thymus, and lymph nodes) of 2D2 male and female mice to be considerably different. These changes in hematopoietic stem cells differentiation profiles and level of lymphocyte proliferation in various organs of 2D2 mice were found to induce the production of IgGs against DNA, myelin basic protein, and myelin oligodendrocyte glycoprotein, increasing the number of autoantibodies hydrolyzing these substrates. We compared the changes of these immunological and biochemical parameters in 2D2 mice with those of mice of two other lines (Th and C57BL/6), also prone to spontaneous development of EAE. Some noticeable and even extreme variations were found in the time-related development of parameters between male and female mice of 2D2, Th, and C57BL/6 lines. Despite some differences, mice of all three lines demonstrated the changes in hematopoietic stem cells profiles, lymphocyte content, and production of catalytic autoantibodies. Given that these changes are harmful to mice, we believe them to cause the development of experimental autoimmune encephalomyelitis.

## 1. Introduction

Multiple sclerosis (MS) is known as a disease of the central nervous system (CNS) involving an increase in T lymphocytes and macrophages. An undeniable reason for multiple sclerosis is still unknown [1]. Several studies suggested myelin disintegration to be important for autoimmune (AI) pathology development. The activated CD4+ myelin-reactive cells are considered as MS mediators [1]. Several latest studies further confirm that B cells and autoantibodies (auto-Abs) against autoantigens of myelin play a significant role in MS pathogenesis [1,2,3]. The increased accumulation of autoantibodies and B cells in the cerebrospinal fluid (CSF), with several specific contraventions in MS patients, provide further evidence for the demyelination involvement [4]. Several animal model analyses indicate that auto-Abs components against myelin participate in Ab-dependent demyelination [3]. Autoantibodies to protein-oligodendrocyte progenitors of cells are supposed to inhibit remyelination by removing or impeding these cells [5].

One of the specific features of AI disease progression was shown to be the appearance of antibodies (abzymes) in the patients’ blood that hydrolyze DNA, RNA, proteins, nucleotides, and oligosaccharides (for review, see [6,7,8,9,10,11,12]). The blood of MS patients was found to contain abzymes that efficiently hydrolyze DNA [13,14], RNA, specific microRNA [15], the myelin basic protein [16,17,18], five histones [19], and oligosaccharides [20,21]. Interestingly, IgG antibodies from the cerebrospinal fluids of MS patients were 30–60 times more active in hydrolysis of MBP, DNA, and oligosaccharides than from the serum of the same patients [22,23,24].

Autoimmune diseases (AIDs) were suggested at first to appear as a result of hematopoietic stem cell (HSC) defects [25]. There are several various mice models of EAE, including C57BL/6, mimicking a particular MS facet of humans (for a review, see [26,27,28]). These mice are characterized by specific B and T lymphocyte responses to antigens [26,27,28]. Another model corresponding to autoimmunity in EAE is the Th mouse model obtained by crossing transgenic mice with a particular myelin-specific T-cell receptor (TCR) and specific for myelin-specific Abs heavy chain knock-in mice [29]. Another MS model is 2D2 TCR (TCR^MOG^) mice expressing a MOG-specific T cell receptor [26,30,31,32,33].

C57BL/6 [34,35,36,37] and Th [38,39] mice were analyzed to study possible mechanisms of spontaneous (in untreated mice) and antigen-accelerated/stimulated EAE development. C57BL/6 and Th mice were shown to exhibit very slow spontaneous and significantly accelerated MOG-induced EAE development [34,35,36,37,38,39]. Some typical indicators of EAE development (optic neuritis and other clinical or histological evidence) appear relatively early only after treating these mice with MOG.

The spontaneous and antigen-induced evolution of EAE in AI-prone C57BL/6 and Th mice [34,35,36,37,38,39], as well as systemic lupus erythematosus (SLE) in MRL-lpr/lpr mice [40,41,42], was demonstrated to occur from the immune system-specific readjustment of bone marrow (BM) HSCs. The bone marrow immune system defects are associated with specific changes in the profile of BM HSCs differentiation [BFU-E (erythroid burst-forming unit (early erythroid colonies)], CFU-E [erythroid burst-forming unit (late erythroid colonies)], CFU-GM [granulocyte-macrophage colony-forming units], and CFU-GEMM [granulocyte-erythroid-megakaryocyte-macrophage colony-forming units], T and B lymphocytes combined with the synthesis of specific catalytic auto-Abs (abzymes) degrading DNAs, RNAs, oligosaccharides, peptides, and proteins. The production of auto-Abs with catalytic activities was revealed as the earliest and statistically reliable, undoubtedly important indicator of the onset of several autoimmune diseases in humans and AIDs-prone mice (for review, see [7,8,9,10,11,12,34,35,36,37,38,39,40,41,42]). Enzymatic activities of abzymes are reliably detectable before the appearance of specific known external and biochemical markers of various autoimmune diseases at the pre-disease stage [7,8,9,10,11,12,34,35,36,37,38,39,40,41,42]. At the pre-disease stage and onset of various AIDs, the concentrations of different auto-Abs most often correspond to the parameter ranges typical for healthy humans and experimental animals. The emergence of catalytic antibodies may authentically testify to the appearance of AIDs, while the increase in their enzymatic activities is associated with the development of profound pathologies [7,8,9,10,11,12,34,35,36,37,38,39,40,41,42]. Nevertheless, the onset of AIDs may be mediated by plural mechanisms leading to self-tolerance destruction. Understanding all possible additional comorbid disease mechanisms is essential for explaining how MS and EAE develop.

As mentioned above, there are several EAE mouse models of human multiple sclerosis: C57BL/6 [26,27,28], Th [29], and 2D2 TCR (TCR^MOG^) mice [30,31,32,33]. Previously, we analyzed several parameters characterizing the onset of EAE in C57BL/6 using only males [34,35,36,37], while in the case of Th mice, both male and female mice were used [38,39]. It was interesting to carry out a detailed comparison of different parameters characterizing the spontaneous EAE in C57BL/6, Th, and 2D2 mice. MS is known to be detected more often in women than in men [43]. Usually, only about 30% of patients are men with more severe pathology that is less responsive to therapy. Moreover, the disease is usually observed rather late in men.

Some significant differences in immunoregulation in men and women can occur not only in the case of stem cell differentiation and production of auto-abzymes. For example, in patients with MS, miR-223-3p and miR-379-5p were amplified only in men [44,45]. Enhanced expression of miR-379, miR-127-3p, miR-431, miR-381, miR-656, miR-376c, and miR-410 was also detected only in men but not in women with relapsing-remitting MS [44,45]. Thus, the development of MS in women and men can differ in terms of various changes in any other parameters. Therefore, it was interesting to compare the development of EAE in 2D2 male and female mice by analyzing cell differentiation and abzyme catalytic activity. The development of MS in different people occurs in slightly different ways, however, ultimately leading to similar medical indicators of the disease. Therefore, understanding possible differences and similarities in the development of MS in humans is only possible by analyzing several different models of EAE-prone mice.

Here, for the first time, we have analyzed different indicators showing the development of EAE in 2D2 TCR (TCR^MOG^) mice. The differentiation profiles of BFU-E, CFU-E, CFU-GM, and CFU-GEMM colonies and T and B cells of 2D2 mice BM at the spontaneous progress of EAE were examined. In addition, we estimated the changes in proliferation profiles of total lymphocytes, B, T, CD4, and CD8 cells in various organs. The concentrations of antibodies against DNA, MBP, and MOG and relative activities of IgGs hydrolysis these substrates were estimated at different phases of EAE development in male and female 2D2 mice. In addition, we have compared the data obtained in 2D2 mice with the results previously obtained in C57BL/6 and Th mice.

## 2. Results

### 2.1. Choosing a Model for Analyzing EAE Development

B and T lymphocytes play vital roles in human MS pathogenesis [1]. B cells secreting Abs ensure the humoral immunity of the immune system [46]. In contrast, natural killer and T lymphocytes in the BM and mature B cells possess membrane receptors binding various antigens. B and T lymphocyte responses characterize spontaneous and MOG-accelerated EAE in C57BL/6 mice (often used to model human MS) [26,27,28]. In Th mice prepossessed to spontaneous EAE, B cells recognize MOG, and plasma cells constitutively synthesize pathogenic anti-MOG Abs [29,32]. Th mice can be referred to as mice with a specific B- and T-cell response (T cell-mediated CNS AID) because Abs against MOG are required to accelerate spontaneous EAE by opsonizing the endogenous antigen in the absence of B lymphocytes when mice endogenously contain MOG-recognizing T cells [32]. 2D2 TCR (TCR^MOG^) mice express a MOG-specific T cell receptor (TCR) [30,31,32,33]. T transgenic cells are functionally competent. The main part of 2D2 mice thymocytes shows an increased receptor level, indicating efficient selection of transgenic T cells.

Spontaneous EAE in C57BL/6 and Th mice is associated with unhurried changes in differentiation profiles of HSCs and an increased rate of lymphocyte proliferation in various organs [34,35,36,37,38,39]. These processes in C57BL/6 and Th mice are accompanied by the appearance of auto-Abs against MBP, MOG, and DNA [34,35,36,37,38,39]. Treating C57BL/6 and Th mice with MOG-peptide resulted in a strong speed-up of EAE development, i.e., the appearance of the acute phase at days 17–20 after the treatment [34,35,36,37,38,39]. During commencement-onset (days 6–7) and the sharp-acute phase (days 17–20) of EAE, very significant changes in the HSC profiles of cell differentiation were observed in parallel with an increased level of B and T cell proliferation as well as the synthesis of Abs against MBP, DNA, and MOG hydrolysis these antigens-substrates [34,35,36,37,38,39]. Auto-abzymes degrading MBP, MOG, and DNA are considered hazardous for mammals. Auto-Abs-abzymes that hydrolyze DNA penetrate through cellular and nuclear membranes and destroy chromatin DNA, resulting in cell apoptosis [47,48,49]. This leads to increased blood concentrations of DNA-histone complexes, the main mammalian antigens that stimulate the production of Abs against histones and DNAs [7,8,9,10,11,12,47,48,49]. Autoantibodies that degrade MBP and MOG hydrolyze them in the nerve tissue membranes, disrupting nerve impulses [7,8,9,10,11,12].

We have previously analyzed several parameters characterizing the development of EAE in C57BL/6 using only males [34,35,36,37], while in the case of Th mice, we studied both males and females [38,39]. Therefore, it was important to compare the development of EAE in male and female 2D2 mice with C57BL/6 and Th mice by analyzing cell differentiation and catalytic activities of abzymes.

2D2 TCR (TCR^MOG^) mice were received from Westfälische Wilhelms-Universität, Department of Neurology (Münster, Germany). Before starting the work, according to the described properties of this line [30,31,32,33], we checked that 2D2 mice, received during ≥1 year, developed all typical characteristics of EAE, including spontaneous optic neuritis. However, the unexpected development of EAE is a long-term process that leads to severe pathology, but the first weak clinical markers do not appear until 6–7 months of their life, with optic neuritis indicators appearing within ≥1 year. At the same time, we have shown in the example of C57BL/6 and Th mice that the changes in the differentiation profile of BM stem cells and the appearance of catalytic antibodies that are dangerous for mice are detected as early as at three months of these mice’ life. Considering all these facts, it seemed interesting to analyze the initial stages of EAE development in 2D2 mice, eventually leading to the development of severe pathology.

### 2.2. Proteinuria in Different Mice

The development of AIDs in autoimmune humans and mice was shown to correlate well with the increase in proteinuria (2.0–3.0 mg/mL of protein in urine) [34,35,36,37,38,39,40,41,42]. Non-autoimmune mice (BALB and CBA) demonstrated no proteinuria for at least one year (0.1–0.12 mg/mL) [34,35,36,37,38,39,40,41,42]. Healthy autoimmune-pone MRL-lpr/lpr mice were characterized by low proteinuria (0.36–0.39 mg/mL) before the spontaneous development of deep SLE [40,41,42]. However, autoimmune C57BL/6 mice demonstrated a significantly higher level of proteinuria (up to 10–12 mg/mL) at two-three months of age [50]. According to our experimental data, control C57BL/6 mice demonstrated a urine protein concentration of 7.2 ± 0.8 mg/mL at three months of age, and during 40–45 days of spontaneous development of EAE, it increased to 12 ± 0.8 mg/mL (Figure 1) [34,35,36,37].

Three-month-old male Th mice also demonstrated a high level of proteinuria (6.3 ± 1.5 mg/mL), which increased to 12.0 ± 0.9 mg/mL during 70 days (Figure 1) [38,39].

Proteinuria in 2D2 males at zero time (3 months) was nearly the same (6.6 ± 1.3 mg/mL) as in Th males but increased during 77 days only to 8.1 ± 1.0 mg/mL (Figure 1). A completely different result was observed in 2D2 Th female mice. The protein concentration in the urine of Th and 2D2 female mice was approximately 2.3 and 2.8 fold lower at zero time (2.7 and 2.4 mg/mL) than in male mice and changed slightly with time (Figure 1). Thus, this parameter of EAE development in males and females was found to be significantly different, suggesting that changes leading to the development of EAE in females do not result in a strong increase in proteinuria compared to male mice.

### 2.3. Hematopoietic Progenitor Cell Formation

As previously reported, mice not predisposed to autoimmune diseases did not exhibit any noticeable changes in the profile of stem cell differentiation for at least one year [40,41,42]. In SLE-prone MRL-lpr/lpr mice, the spontaneous development of pathology led to changes in the differentiation profiles starting from three months, with all mice showing typical symptoms of the disease at seven months [40,41,42]. The changes in stem cell differentiation profiles of three-month-old 2D2 mice (seven female and seven male mice) were examined. At three months of age, the relative number of BFU colonies in 2D2 male mice was 2.3 times higher than in female mice, with a maximum difference observed at day 15 (7 times) and day 22 (5.4 times); *p* < 0.05 (Figure 2A).

Three strains of three-month-old EAE-prone mice differed much in the relative number of BFU colonies and the time of spontaneous development of this pathology (Figure 2A). At three months of age, the relative number of BFU colonies in male 2D2 mice was 48.4 and 31.7 times less than in Th and C57BL/6 males, respectively. The difference in the number of these colonies in male and female of 2D2 and Th mice decreased greatly by day 50–70 but remained significant compared with C57BL/6 males (~3.0–5.7-fold; *p* < 0.05) (Figure 2A). At the beginning of the experiment, the number of BFU cells in 2D2 females was 10.5 times lower than in female Th mice.

The relative content of CFU-E colonies in female 2D2 mice was only 1.2 times higher than in male 2D2 mice (*p* > 0.05) (Figure 2B). Such a slight difference was observed up to 50 days, with a maximum difference at day 15 (2.4 times, *p* < 0.05). In the beginning, the relative number of CFU-E colonies in male 2D2 and Th mice did not differ significantly—only 1.9 times, and the time-related difference was 2-fold, with a maximum difference of 3-fold during 7–10 days (*p* < 0.05). The differences between CFU-E colonies in Th and 2D2 female mice demonstrated a completely different situation. The difference in the number of CFU-E colonies in three-month-old female Th and 2D2 mice was about 1.4 times, but after the development of EAE in Th by day 52, the number of CFU-E cells increased 3.3 times and became ~6 times (*p* < 0.05) higher than in female 2D2 mice (Figure 2B). At three months of age, the relative content of CFU-E colonies in male 2D2 mice was 2.7 times lower than in C57BL/6 males (Figure 2B). However, the number of these colonies in C57BL/6 males grew faster in time, and by day 42, it became six times higher than in 2D2 mice (*p* > 0.05).

Figure 2C demonstrates the changes in CFU-GM colonies in male and female 2D2 mice. At zero time, the relative number of CFU-GM colonies in male 2D2 mice was 1.7 times higher than in female 2D2 mice. Later, a steady increase in the number of these colonies was observed, being faster in males than in female mice (Figure 2C). At day 43, a maximum 2.0-fold difference in their number was observed (*p* < 0.05). The relative number of CFU-GM colonies in three-month-old male 2D2 and Th mice was almost the same. However, Th males showed a slight change in these cells over time, while male 2D2 mice showed a substantial change. This led to the fact that by day 43, the content of CFU-GM colonies in the BM of male 2D2 mice was about 8.4 times higher than in male Th mice, with a more significant difference of about 14 times for female mice (*p* < 0.05). Three-month-old mice showed the most significant 7.7-fold difference in the content of CFU-GM colonies between male 2D2 mice and male C57BL/6 mice (Figure 2C). However, after 60–70 days, the number of these colonies in C57BL/6 males increased only 1.4-fold, while in 2D2 males, it increased 8.4-fold (Figure 2C).

At the beginning of the experiment, the relative number of CFU-GEMM colonies of male and female 2D2 mice was approximately the same (Figure 2D). Then, the growth of these cells in males was faster than in female mice, and on day 15, their content in males was 5.2 times (*p* < 0.05) higher than in females. The relative content of CFU-GEMM colonies in 2D2 male mice at zero time was 3.4–5.8 times lower than in C57BL/6 males and Th males or females (Figure 2D). Th and C57BL/6 males showed a slight decrease in these cells over time, while female Th mice demonstrated a slight increase. Finally, the number of these cells at day 43 in male and female 2D2 mice was significantly higher than that in C57BL/6 and Th mice.

The total relative content of B cells in the BM of males was 1.3-fold (*p* > 0.05) higher than in females at zero time (Figure 2E). By day 20, a marked increase in their number was observed, followed by a strong decrease by day 97. Interestingly, in 3-month-old mice, the relative content of B lymphocytes in male and female 2D2 mice was ~1.5 times higher than in male and female Th mice. With Th mice showing a constant decrease in B-lymphocyte content over time, 2D2 mice showed an increase in their content up to 20 days. (Figure 2E).

The relative T-cell content of three-month-old Th males was 8.2-fold higher than that of 2D2 males (Figure 2F). At the same time, the spontaneous development of EAE in 2D2 males demonstrated a 4.9-fold increase by day 70, and in Th males—a 1.8-fold decrease, making their values for the two lines of male mice comparable.

At the beginning of the experiment, the relative content of T cells in Th females was 2.1 times higher than in 2D2 females. During the spontaneous development of EAE in female 2D2 and Th mice within 70 days, a gradual and nearly similar 1.4–1.6-fold increase in their number was observed (Figure 2F).

Unexpectedly, contrasting changes in the content of T cells in the BM of male 2D2 and Th mice occurred over time (Figure 2F). Their relative number for 2D2 and Th females differed significantly, but the variation was almost the same over time (Figure 2F). Thus, the differentiation profiles of BM stem cells in 2D2 and Th males and females during the spontaneous development of EAE have significant differences. Although the relative content of BFU-E, CFU-E, CFU-GM, and CFU-GEMM cells in 2D2 females and males differed significantly at zero time, the character of their changes during the spontaneous development of EAE over time was similar in some respects. However, significant differences were found in the relative number and pattern of changes in all six types of BM cells in 2D2 mice compared to those in Th and C57BL/6 mice (Figure 2).

### 2.4. The Relative Content of B and T Cells in the Organs of Mice

At the beginning of the experiments, we estimated the relative average content of total B, T, and CD4 and CD8 cells in multiple organs of seven males and seven females (Table 1). The average content of B cells in the blood and various organs of 2D2 mice was significantly different and decreased in the following order: lymph nodes > thymus > spleen ≈ blood > bone marrow (Table 1). The relative number of T cells in the blood and various organs changed in a different order: spleen > blood > bone marrow lymph nodes > thymus (Table 1).

Figure 2E shows the changes in relative amounts of B lymphocytes in the BM. Figure 3 presents the changes in four other organs of 2D2 mice.

The relative content of B cells in the blood of 2D2 males was ~1.2-fold higher than that in females (*p* > 0.05), with an increase in their number by about 1.2 times in both cases within 40 days. Interestingly, in the case of 3-month-old Th mice, the relative number of blood B lymphocytes was significantly less: 2- and 1.5-fold in males and females, respectively. However, the relative rise in their content over time in 2D2 and Th mice is quite comparable (Figure 3A).

The patterns of changes in the content of B lymphocytes in the lymph nodes (Figure 3B) and spleen (Figure 3C) of 2D2 males and females are very complex. In the lymph nodes at the very beginning of the development of EAE, there was a sharp increase in B lymphocytes (7 days), followed by a significant decrease by day 20. Then (>30 days), a more substantial rise in the number of B-lymphocytes in males than in females was detected (Figure 3B). Notably, male and female Th mice demonstrated a slight reduction in the content of B lymphocytes by day 7 and then an increase in their number, as compared to 2D2 mice.

Somewhat different changes in the content of B-lymphocytes were observed in the spleen of 2D2 mice (Figure 3C). At first, there was a noticeable decrease in their number in 2D2 mice by day 7, such as in Th mice. After 14–20 days, a strong increase in the relative content of B cells happened. By day 41, both male and female 2D2 mice demonstrated a dramatic decrease in the number of B cells, followed by a substantial increase in their relative number up to day 97 (Figure 2C). It is quite curious that Th mice also showed a decrease in the number of B-lymphocytes in the spleen by day 7 and then a significant increase within 14–20 days (Figure 2C).

The thymus of Th males and females initially contained approximately the same number of B cells, and during spontaneous development of EAE, a smooth rise in their number was observed (Figure 3D). At zero time, the relative number of B lymphocytes in males was 2.9-fold higher than in 2D2 females (*p* < 0.05), but the pattern of their number increasing for 77 days was very similar (Figure 3D).

The dependencies of the changes in the relative content of B lymphocytes in various organs of 2D2 mice were different in general (Figure 3). In addition, the change in the content of B lymphocytes in various organs of 2D2 mice differed significantly from that in Th mice.

### 2.5. The Content of T Lymphocytes in the Organs of Mice

The patterns of changes in the relative content of T cells in the BM of Th males and females were directly opposite (Figure 2F). However, in the BM, the patterns of changes during spontaneous development of EAE in 2D2 male and female mice were very similar (Figure 2F). Time-related changes in the relative content of T cells in 2D2 and Th mice in the blood and lymph nodes were approximately opposite. (Figure 4A).

While female and male 2D2 mice showed a strong decrease in the content of T cells in the blood, Th mice exhibited a sharp increase in T cells’ number in the early period (7 days), followed later by a strong decrease (Figure 4A). In 2D2 males and females, the content of T lymphocytes in the lymph nodes increased up to 20 days, followed by a significant decrease (Figure 4B). At the same time, Th mice demonstrated a slight change in the number of T lymphocytes during the slow spontaneous development of EAE.

The spontaneous development of EAE led to a decrease in the content of T cells in the spleen of 2D2 and Th mice (Figure 4C). The patterns of changes in the number of T cells in Th females and males are very similar. However, 2D2 males and females exhibited complex dependencies in T-cell number decrease over time. Notably, the relative content of T cells in the thymus of three-month-old 2D2 mice was approximately 4.3–4.7 times higher than that of Th males and females (Figure 3D). However, during the spontaneous development of EAE in both cases, a slow rise in their number was observed.

Thus, during the spontaneous development of EAE in 2D2 mice, significant differences were observed in changes in the relative numbers of B and T lymphocytes in the analyzed organs of females and males. Despite both being prone to spontaneous development of EAE, 2D2 and Th lines of mice showed somewhat different patterns of changes in the content of B and T cells in various organs.

### 2.6. The Relative Content of CD4 and CD8 Lymphocytes

The initial relative number and character of time-related changes in the total content of CD4 and CD8 cells in blood and various organs were different (Table 1); initial CD4 cells: spleen > thymus > blood > lymph nodes > bone marrow; initial CD8 cells: lymph nodes > thymus > spleen > blood > bone marrow.

In most organs of 2D2 mice, the relative amount of CD4 cells was noticeably higher than that of CD8 lymphocytes, with their ratio varying from 1.7 to 41.5 (Table 1). Figure 2E shows the changes in the relative number of B-lymphocytes in the BM, and Figure 3 presents the changes in the other four organs of 2D2 mice.

Only in the case of lymph nodes, the relative content of CD8 exceeds the CD4 lymphocyte content in male and female mice by 24.7–52.7 times (Table 1). The relative average CD4 cell content in various organs was predominantly higher in males than in females (1.1–4.0-fold), except for the BM having 1.5-fold more CD8 cells than in male mice. Roughly the same situation was observed with CD8-lymphocytes, with their number being 1.3–2.4 times higher in male than in female mice, except for lymph nodes and thymus, where their average values were 1.1–1.2 times (*p* > 0.05) lower in males than in female mice (Table 1).

During the spontaneous development of EAE, a slow rise in the number of CD4 cells was observed both in 2D2 males (2.3-fold) and females (1.3-fold) (Figure 5A).

The increase in the CD4 cells also occurred in the lymph nodes of male (4.5-fold) and female (5.6-fold) mice (Figure 5B). In blood and other organs, there was a downward tendency of CD4 lymphocyte content over time (-fold): spleen (~1.6 males and females), thymus (1.6 males and 2.2 females; Figure 5D), blood (2.1 males and 3.5 females; Figure 5E). Interestingly, the patterns of changes in the relative content of CD4 lymphocytes in organs of 2D2 mice differed in almost all cases from those for the Th line of EAE mice. Although 2D2 mice showed an increase in the number of CD4 cells in the BM and lymph nodes during the spontaneous development of EAE, Th mice demonstrated a significant reduction in their number (Figure 5A,B). Only the spleen demonstrated a decrease in CD4 cells in both 2D2 and Th mice (Figure 5C). A decreased number of CD4 cells in the thymus of 2D2 mice corresponds to their increased number in Th mice (Figure 5D).

2D2 mice showed a gradual decrease in the content of CD4 cells in the blood, while Th mice first showed a sharp increase in their number up to day 20, followed by a relatively weak decrease (Figure 5E). The pattern of the CD8-lymphocyte rises in the BM of male (3.6-fold) and female (6.5-fold) 2D2 mice (Figure 6A) was very similar to that of CD4 cells (Figure 5A).

A smooth increase in the number of CD8 lymphocytes was observed in the blood and spleen of 2D2 mice over time. There was a 3.0–3.3-fold increase in the number of lymphocytes in the blood of males and females (Figure 6B). In the spleen, females showed a higher increase in CD8 number (4.6-fold) than males (2.6-fold) (Figure 6C).

In contrast with the organs mentioned above, the lymph nodes and the thymus of 2D2 mice showed complex change patterns in CD8 lymphocytes over time. In the first 20–30 days of the experiment, a remarkable increase in the CD8 lymphocyte number was observed, followed by a decrease (Figure 6D,E). Interestingly, the pattern of changes in CD8 cells in 2D2 mice differed from that in Th mice, as was the case with CD4 lymphocytes (Figure 6). Of particular interest are the differences observed for the time changes in the relative content of CD8 cells in the BM of 2D2 and Th mice. While female mice of both strains demonstrated increased CD8 lymphocyte content over time, male mice showed increased content in the 2D2 line and a substantial decrease in Th mice.

Reverse patterns of changes in the content of CD8 cells were observed in the spleen of female 2D2 and Th mice (Figure 6C). Only the lymph nodes and thymus of 2D2 mice showed a noticeable increase in CD8 lymphocytes in 10–30 days, while in Th mice, there was an insignificant change in the content of these cells (Figure 6D,E). Thus, during the spontaneous development of EAE, quite different changes in the relative content of B, T, CD4, and CD8 lymphocytes can occur in various organs of 2D2 mice in general, as well as in males and females of this line. In addition, the changes in the content of these cells in organs in two lines of 2D2 and Th mice were found to differ significantly.

### 2.7. Intrinsic Nature of the Catalytic Activity of Antibodies

Electrophoretically homogeneous IgGs from individual 2D2 mice were isolated by affinity chromatography of blood sera components on Protein G-Sepharose while removing nonspecifically bound proteins as described in [34,35,36,37,38,39,40,41,42]. Then, these IgG preparations were additionally purified by FPLC gel filtration. IgG homogeneity was shown using SDS-PAGE with silver staining and equal amounts of 2D2 mice IgGs (IgG_mix_) (Figure 7A) similar to [34,35,36,37,38,39,40,41,42]. Using rigorous criteria, DNA-, MOG-, MBP-, and histone-hydrolyzing activities were shown to be intrinsic properties of IgG_mix_ containing no co-purified canonical enzymes. The most important criteria include: (i) electrophoretic homogeneity of IgGs (Figure 7A); (ii) FPLC gel filtration of IgG_mix_ using acidic shock (pH 2.6) did not lead to a disappearance of the activities, and the peaks of DNase and protease activities (hydrolysis of MBP, MOG, and histones) tracked exactly with IgGs (Appendix A); (iii) complete absorption of all activities by Sepharose bearing Abs against mouse IgGs led to catalytic activities disappearing from solution, followed by their being eluted from the sorbent with acidic buffer (pH 2.6; Appendix A).

Described in the literature [6,7,8,9,10,11,12] is one criterion, the fulfillment of which also makes the rest of the criteria be met.

SDS destroys any complex of proteins, while the electrophoretic mobilities for canonical DNases and proteases (28–32 kDa) are significantly higher than for IgGs (150 kDa). It was shown that after SDS-PAGE, the positions of DNA-, MOG-, MBP-, and histones-hydrolyzing activities of gel fragments corresponded to intact IgGs. No bands corresponding to other proteins or their possible complexes with antibodies (Figure 7A) and no other peaks of catalytic activities (Figure 7B) were detected. Thus, the detection of DNase-, MOG-, MBP- and histone-hydrolyzing activities only in IgG_mix_ peaks after Abs gel filtration under conditions of immune complex destruction, affinity chromatography on Sepharose bearing immobilized rabbit IgGs against mouse IgGs, and detection of four IgG_mix_ activities only in the gel fragments corresponding to intact IgGs (Figure 7B) provided direct evidence of mouse IgGs possessing DNA-, MOG-, MBP-, and histone-hydrolyzing activities.

### 2.8. Abs against Proteins and DNA

The sera of healthy people and animals tend to contain autoantibodies against DNA and different proteins in low concentration, which are likely to increase dramatically in various types of AIDS [7,8,9,10,11,12]. We estimated the concentration of Abs against DNA, MBP, MOG, and histones using the obtained homogeneous IgG preparations of 2D2 mice and compared these values with those for Th and C57BL/6 mice. The relative average concentration of IgGs against DNA (at three months of age) corresponding to 2D2 males was 4-fold higher than for females (Figure 8A).

The concentration of anti-DNA IgGs in 3-month-old C57BL/6 males exceeded that of 2D2 and Th males by 2.4–3.5 times and did not change much with time, following a complex curve (Figure 8A) [34,35,36,37]. While females showed a constant time-related decrease in concentration, males demonstrated a significant increase after day 20. Interestingly, in Th males and females, the concentration at zero time was comparable to that of 2D2 females. However, then, in contrast with 2D2 mice, it dramatically increased in males and females (3.8–5.4-fold) during 73 days of the experiment (Figure 8A) [38,39].

In female and male 2D2 mice, a slight increase was observed in the concentration of IgG antibodies against MBP by the days 7–14, followed by a noticeable decrease (Figure 8B). Female and male Th mice demonstrated ~3-fold higher concentration at 3 months, and then it decreased ~2 times for 73 days [38,39]. At zero time, the highest concentration of IgGs against MBP is observed in C57BL/6 male mice and its substantial increase in the time (Figure 8B).

The relative amount of anti-MOG Abs in the total IgG preparations of 2D2 females was comparable to that in Th male and female mice, and in all cases, there was a slight increase in their concentration (Figure 8C). However, the initial concentration of anti-MOG IgGs in 2D2 males was ~2.5 times higher than in female mice. In 2D2 males, there was a rise in the concentration of these IgGs up to 20 days, followed by a subsequent remarkable decrease (Figure 8C). C57BL/6 male mice demonstrated a constant solid increase in the concentration of anti-MOG IgGs over time [34,35,36,37].

The average relative content of IgGs against the sum of five histones in 2D2 males was estimated to be about two times higher than in female mice (Figure 8D). At the same time, the patterns of time-dependent changes in the relative content of IgGs against histones proved very similar: a slight increase during day 7 to day 14 with a subsequent decrease (Figure 8D). In female and male Th mice, the relative concentration of antibodies against histones did not change significantly over time. The initial concentration of IgGs against histones in C57BL/6 male mice at zero time was to some extent comparable to that in female 2D2 and male and female Th mice, although increasing very strongly during spontaneous development of EAE (Figure 8D).

Thus, despite all three lines of mice being predisposed to spontaneous development of EAE, the change in the relative concentration of autoantibodies against some antigens in them was very different. In addition, in some cases, comparable concentrations of antibodies against some antigens for males and females were observed, including to some extent similar patterns of their change over time, with some antigens and mice showing significant differences in these parameters.

### 2.9. Changes in Catalytic Activities of IgGs

The appearance of antibodies with catalytic activities proves to be a very specific and earliest marker of the onset and subsequent development of various autoimmune diseases [7,8,9,10,11,12]. Antibodies of non-autoimmune healthy mice BALB and CBA and healthy humans usually possess no hydrolytic activities [7,8,9,10,11,12]. The blood of diseased MRL-lpr/lpr SLE-prone mice contains autoantibodies hydrolyzing DNAs, ATP, and oligosaccharides [41,42,43]. C57BL/6 EAE mice during spontaneous development of the pathology demonstrated a relatively slow and near-linear 6.8-fold increase in DNase activity by day 73 of spontaneous development of EAE (Figure 9A) [34,35,36,37].

In this work, the relative activities of IgGs isolated from individual 2D2 mice of experimental groups were evaluated using the standard methods described previously [35,36,37,38,39]. To illustrate, Appendix A shows typical hydrolysis patterns of DNA, MBP, MOG, and five histones in the case of IgGs from the blood of five different 2D2 mice. The relative activity in the hydrolysis of the four substrates was calculated in terms of the decrease in the substrates compared to the controls incubated in the absence of antibodies (see Section 4 and caption to Appendix A).

At three months of age (zero time), DNA-hydrolyzing activity of Th male antibodies was ~2.5 times lower than that of females (Figure 9A) [38,39]. For 73 days, the DNA-hydrolyzing activity of Th male IgGs increased ~11 times, with only the 2.8-fold increase for female Abs activity (Figure 9A). DNase activity of IgGs of 3-month-old male 2D2 mice was 36.4- and 89-fold higher than that of Th and C57BL/6 mice, respectively. Initially, the activity of antibodies of male and female 2D2 mice was comparable, and within 97 days of slow spontaneous development of EAE, under complex but similar dependencies, it increased 4.1–4.8 times (*p* < 0.05) (Figure 9A). At 70–80 days after the beginning of the experiment, DNase activity of antibodies of 2D2 mice was 6.5–27.0 times higher than that of Th and C57BL/6 mice.

At three months of age (zero time), DNA-hydrolyzing activity of Th male antibodies was ~2.5 times lower than that of females (Figure 9A) [38,39]. For 73 days, the DNA-hydrolyzing activity of Th male IgGs increased ~11 times versus only a 2.8-fold increase of female Abs activity (Figure 9A). DNase activity of 3-month-old male 2D2 mice IgGs was 36.4- and 89-fold higher than that of Th and C57BL/6 mice, respectively. Initially, the activity of antibodies of male and female 2D2 mice was comparable, and within 97 days of slow spontaneous development of EAE, under complex but similar dependencies, it increased 4.1–4.8 times (*p* < 0.05) (Figure 9A). By days 70–80 after the beginning of the experiment, DNase activity of 2D2 mice antibodies was 6.5–27.0 times higher than that of Th and C57BL/6 mice.

The initial RA of 2D2 mice IgGs in MBP hydrolysis in males and females was almost identical (Figure 9B). The gradual increase in IgGs activity in 2D2 male and female mice up to 20 days was also nearly the same. Notably, MBP-hydrolyzing activity of antibodies in 3-month-old 2D2 mice was about 2.2-fold higher than in C57BL/6 and 9–10 times higher than in male and female Th mice (Figure 9B). All three lines of mice showed an increase in MBP-hydrolyzing activity during their slow spontaneous development of EAE disease, with the maximum 8–9-fold rise in antibody activity demonstrated by Th mice.

In contrast to the MBP-hydrolyzing activity, the MOG-hydrolyzing antibody activity of male and female 2D2 mice was 2.4–2.7 times lower than that of C57BL/6 mice (Figure 9C). However, there was a substantial slowdown in the rise of MOG-hydrolysis activity after 40 days in sera of female compared to 2D2 male mice (Figure 9C).

### 2.10. Time-Dependent Changes in IgG-Dependent Hydrolysis of Individual Histones

The blood of humans and animals with AIDs was previously shown to contain autoantibodies against five histones, which hydrolyze these histones [19]. Here, we analyzed the relative activity of 2D2 mouse IgG preparations in the cleavage of individual histones during the development of EAE. Typical examples of relative activity analysis by SDS-PAGE using five antibody preparations are shown in Appendix A. The decrease in the relative protease activity in the hydrolysis of five histones was different in male and female mice, with male mice showing H1 > H2A ≈ H3 > H4 > H2B and female mice demonstrating H3 > H1 > H2A ≈ H4 ≈ H2B (Table 2).

The highest initial activity was observed in the hydrolysis of H3 histone by antibodies of female mice, with the lowest one being in the hydrolysis of H2B by IgGs of male mice (Table 2). The most considerable difference (2.2-fold) in average values of histone hydrolysis by antibodies of male and female mice was observed for H3 histone. The maximum difference (8.3-fold) in the hydrolysis of different histones was found for H3 of females and H2B for male mice IgGs. Despite the significant difference in the efficiency of hydrolysis of five histones by Abs of males and females at three months of age, all cases of the slow spontaneous development of EAE showed an increase in activity up to 20–40 days followed by a decrease (Figure 10). The maximum increase in activity (8-fold) was found in the case of H3 histone hydrolysis by male mouse IgGs at day 8 (Figure 10D).

To summarize, some perceptible or significant differences have been found to be evident for males and females not only in the dependencies of time-related changes in differentiation profiles of stem cells (Figure 1) and patterns of changes in lymphocyte proliferation in various organs (Figure 2, Figure 3, Figure 4, Figure 5 and Figure 6) but also in relative concentrations of Abs against DNA, MBP, MOG, and histones and IgG activities in their hydrolysis at three months of age and different stages of slow EAE development (Figure 9 and Figure 10).

## 3. Discussion

Different autoimmune diseases were shown to start due to self-tolerance breakdown via multiple mechanisms. EAE-prone C57BL/6, Th, 2D2, and SLE-prone MRL-lpr/lpr mice are predisposed to the spontaneous development of autoimmune pathologies [34,35,36,37,38,39,40,41,42]. The spontaneous slow appearance of EAE and SLE in these mice can be significantly accelerated by their immunization with DNA and MOG [34,35,36,37,38,39,40,41,42]. Interestingly, the development of EAE in C57BL/6 and SLE in MRL-lpr/lpr mice leads to somewhat similar changes in the profiles of HSCs differentiation at their onset and production of various abzymes [8,9,10,11,12].

Given that activation of myelin-reactive T and B cells is critical for the pathogenesis of MS in mammals [1,2,3], it was interesting to analyze and compare the changes of different parameters characterizing the slow spontaneous development of EAE in C57BL/6, Th, and 2D2 mice. In addition, with women being more predisposed to MS than men [43], it seemed important to compare the patterns of changes in several parameters during spontaneous development EAE in males and females of 2D2 and Th mice.

It is worth noting that the type of changes in the relative number of all six precursors of blood cells in the BM of 2D2 mice during slow spontaneous development of EAE was only to some extent similar in males and females (Figure 2). The main differences were observed primarily in the relative content of these cells in 3-month-old males and females (Table 1) and the amplitude of their changes over time (Figure 2).

The most remarkable differences in changes in the relative content of various cells during the development of EAE in 2D2 mice were observed for BFU-E, CFU-GEMM, T, and B lymphocytes (Figure 2). The relative content of BFU-E cells increased in 2D2 but decreased in Th mice. The strong increase in CFU-GEMM in 2D2 mice was not consistent with their decrease in C57BL/6 and Th mice. Opposite patterns of changes in T lymphocytes in BM were observed in males of 2D2 and Th mice (Figure 2). Overall, the characteristics of changes in the number of B lymphocytes in male and female 2D2 mice proved to some extent similar (Figure 3). The most striking differences in the change over time in the relative content of B cells were observed in the lymph nodes of 2D2 and Th mice (Figure 3B).

The curves characterizing changes in the content of T lymphocytes in 2D2 males and females have a comparable character and differ mainly in the magnitude of the resulting changes (Figure 4). Exactly opposite patterns of time-related changes in the content of T lymphocytes were found in the BM and lymph nodes of 2D2 and Th mice (Figure 2F and Figure 4B).

During the spontaneous development of EAE, all three strains of mice showed increased abzyme activity in DNA hydrolysis, MBP, and MOG (Figure 9). However, the relative activities (RAs) of IgGs of 3-month-old mice were very different for 2D2, Th, and C57BL/6, with these differences ranging from (-fold): 1.2–2.8 (MOG-hydrolyzing activity), 2.2–9.1 (MBP-hydrolyzing activity), 16.6–89 (DNase activity) (Figure 9A). In addition, the increase in the level of RAs during the development of EAE was also very different. The relative activities in the IgG-dependent hydrolysis of all five histones, and individual histones, can be very different in both 2D2 males and females (Table 2). The maximum difference (8.3-fold) in the cleavage of two different histones was observed for H3 (45.4 ± 3.0%, female mice) and H2B (5.5 ± 2.0%, male mice). H2B histone was cleaved by antibodies of females 2.2 times faster than by IgGs of male mice (Table 2). At the same time, the nature of time-related changes in RA for all histones is somewhat similar: an increase in the activity during the first 20–40 days followed by a decrease (Figure 10). However, they were markedly or incredibly different in time-related changes of EAE development.

Of interest is a significant difference in the alteration of some analyzed parameters demonstrated by male compared to female mice in the case of three different strains of EAE-prone mice.

All the data obtained indicate that different strains of mice prone to spontaneous development of EAE with increasing age may exhibit greatly different changes in all parameters, with all of them eventually demonstrating EAE pathology.

## 4. Materials and Methods

### 4.1. Reagents

Superdex 200 HR 10/30 column, Protein G-Sepharose, several proteins, and other different chemicals and reagents were from GE Healthcare (New York, NY, USA) and Sigma-Aldrich (Munich, Germany). The human myelin basic protein of 18.5 kDa was obtained from RCMDT (Moscow, Russia) and MOG_35–55_ from EZBiolab (Heidelberg, Germany). All preparations were free from oligosaccharides, lipids, nucleic acids, and other possible contaminants.

### 4.2. Experimental Animals

Th inbred 2D2 TCR (TCR^MOG^) mice (3 months of age) described earlier in [30,31,32,33] were obtained by us in the special breeding facility for mice at the Institute of Cytology and Genetics (ICG) in special living conditions free of any pathogens. All experiments were carried out in conformity with protocols of the Institute of Cytology and Genetics of the Bioethical Committee (document number 134A of 7 September 2010) in obedience to the humane principles of the European Communities Council Directive 86/609/CEE for working with animals. The Bioethical committee supported our study.

An increase in proteinuria (2.0–3.0 mg/mL concentration of protein in urine) was shown to be a general index of different AIDs development [34,35,36,37,38,39,40,41,42]. The relative proteinuria of mice was analyzed as before [40,41,42].

### 4.3. Bone Marrow Progenitor Cells Analysis in Culture

Samples of bone marrow were obtained from mouse femurs; the ability of bone marrow cells to form colonies was estimated as in [34,35,36,37,38,39,40,41,42]. During spontaneous EAE development, mice were decapitated at various times up to 70 days, but the analysis was carried out immediately after sampling in each case. Per one mouse (2 × 10^4^ cells) were grown in four dishes in the standard conditions using specific mouse cells medium (methylcellulose-based M3434; CanadaStemCell Technologies (Vancouver, Canada). This medium was supplemented with stem cell factor, interleukins IL-3, and IL-6 erythropoietin (EPO). The relative number of CFU-GM, CFU-GEMM, BFU-E, and CFU-E cell colonies on the dishes was calculated after 14 days of incubation at 37 °C (5% CO_2_) in a humidified incubator as in [34,35,36,37,38,39,40,41,42].

### 4.4. Evaluation of Different Lymphocytes in Samples of Different Mouse Tissue

The relative content of B and T lymphocytes in the blood and different organs of mice was determined using flow cytometry. Peripheral blood was obtained by standard decapitation of mice. Sodium citrate was used as an anticoagulant. After cell counting, 500 thousand leukocytes (but not more than 150 μL) were taken for cytometric analysis. Cells were incubated with monoclonal antibodies in the dark for 20 min, and then red blood cells in blood samples were lysed for 20 min using a 10-fold volume of RBC lysis buffer (Biolegend, San Diego, CA, USA). Then, cell samples were centrifuged for 10 min and washed with 500 μL of PBS containing 0.02% EDTA and 1% sodium azide. After centrifugation, 50 μL of PBS was added to the cell pellet and analyzed on a flow cytometer. Lymphocytes were isolated from blood, BM, thymus, lymph nodes, and spleen. Bone marrow was derived by washing the cavity of the femur. Lymph nodes and thymus were carefully homogenized in a glass homogenizer, large particles were removed, and cells were resuspended by their passing using a disposable syringe through a needle. Spleen cells were obtained by washing the organ with a medium-filled syringe through punctures in the stroma of the spleen. This method allows obtaining splenocytes without impurities from the organ stroma. Cells were washed twice by centrifugation with RPMI-1640 medium (5 mL) for 10 min at 1500 rpm. After the second centrifugation, 1 mL of RPMI-1640 medium containing 10 mM HEPES, 10% fetal bovine serum, 0.5 mM 2-mercaptoethanol, 2 mM l-glutamine, 100 μg/mL benzylpenicillin, and 80 μg/mL gentamicin were added to the cell pellet, and the cells were counted. To analyze the relative cell content in extracts of various organs, 500 thousand cells were used in 100 μL of PBS buffer containing 10% fetal bovine serum and the conjugates of different specific monoclonal antibodies. To analyze the relative number of various cells, specific anti-CD45-BV510 (Biolegend, 103138), anti-CD3-FITC (Biolegend, 100204), anti-CD4-PerCP (Biolegend, 100432), anti-CD8-APC (Biolegend, 126614), and anti-CD19-PE (Biolegend, 115508) antibodies were used (Biolegend, San Diego, CA, USA). All stainings were carried out following the manufacturer’s recommendations. Cells were incubated for 20 min with monoclonal Abs and then washed by centrifugation after adding 500 μL of PBS containing 0.02% EDTA and 1% sodium azide. After centrifugation, 50 μL of PBS was added to the cell pellet, and the mixture was used for analysis with the BD FacsVerse flow cytometer (BD Biosciences, SanJose, CA, USA). At least 100,000 events were collected for each sample. Gating was carried out as follows: the total population of lymphocytes was isolated according to size and granularity of the cells, the leukocyte population was determined using the pan-leukocyte marker CD45+, and populations of CD3+ and CD3– leukocytes were isolated. In the CD3+ leukocyte population (T cells), CD4+ and CD8+ T cells were determined, and in the CD3– leukocyte population, the content of CD19+ B cells was estimated.

### 4.5. IgG Purification

Electrophoretically homogeneous mouse IgGs were obtained using first sera proteins affinity chromatography on Protein G-Sepharose and then high-resolution by FPLC gel filtration in highly acidic conditions (pH 2.6) as in [34,35,36,37,38,39,40,41,42]. For protecting IgGs from possible viral and bacterial contamination, the preparations were filtered through Millex membranes (0.1 μm). SDS-PAGE of IgGs was carried out using 4–15% gradient gels and visualized by silver staining [34,35,36,37,38,39,40,41,42].

### 4.6. ELISA of Antibodies against Proteins and DNA

The relative concentrations of Abs against DNA, MBP, MOG, and histones in total polyclonal electrophoretically homogeneous IgG preparations were estimated as in [34,35,36,37,38,39,40,41,42]. After treating immobilized Abs with specific anti-mouse Abs conjugated with horseradish peroxidase, all reaction mixtures were incubated with H_2_O_2_ and then with tetramethylbenzidine. The optical density of the solutions (A_450_) after adding H_2_SO_4_ was determined using the Uniskan II plate reader (MTX Lab System, Vienna, VA, USA) [34,35,36,37,38,39,40,41,42]. The relative A_450_ in the samples analyzed was measured using differences in optical density between experimental and control solutions containing no DNA, MBP, MOG, or histones.

### 4.7. DNA-Hydrolyzing Activity Assay

DNase activity of IgGs was estimated according to [13,14,34,35,36,37,38,39,40,41,42]. The 15–20 μL reaction mixtures contained Tris-HCl (20 mM, pH 7.5), 5 mM MgCl_2_, 1 mM ethylenediaminetetraacetic acid (EDTA) 20 mM NaCl, 20 μg/mL supercoiled (sc) DNA (pBluescript), and 0.01–0.1 mg/mL of IgGs. After reaction mixtures incubation for 3–24 h at 37 °C, products of DNA hydrolysis were analyzed using 0.8% agarose gels electrophoresis. After gel staining with ethidium bromide, they were analyzed by Gel-Pro Analyzer v9.11 (Media Cybernetics, Rockwille, MD, USA).

Typical examples of relative DNase activity analysis using five antibody preparations are shown in Appendix A.

The relative DNase activity (RA) was calculated from the relative amount of intact scDNA and its relaxed form. After its incubation with and without IgGs, all initial rates of scDNA hydrolysis were estimated using linear parts of the time dependencies (20–40% of scDNA hydrolysis) and dependencies on the concentrations of IgGs. A complete transition of scDNA to its relaxed form was taken for 100% of the enzymatic activity. Using the reaction conditions of the pseudo-first-order and the linear parts of the dependencies made it possible for comparison to recalculate the relative activities to the same conditions. The RA values (% of the DNA hydrolysis) were finally recalculated to the same standard time, and IgGs concentration in the case of Abs corresponding to 2D2 and used for comparison previously obtained data for C57BL/6 [34,35,36,37], and Th [38,39] mice: percent of DNA hydrolysis in the presence of 0.01 mg/mL IgGs.

### 4.8. Protease Activity Assay

Mixtures (10–50 μL) for protease activity assay contained Tris-HCl buffer (20 mM, pH 7.5), 0.8–1.0 mg/mL of proteins (MOG, MBP, or 5 histones), and 0.001–0.2 mg/mL of IgG preparations as in [34,35,36,37,38,39,40,41,42]. After mixtures incubation for 5–24 h (37 °C), the cleavage products of MOG, MBP and histones were analyzed using SDS-PAGE 3–15% gradient or 12% gels following Coomassie R250 staining. Typical examples of relative protease activity analysis using five antibody preparations are shown in Appendix A. After gel scanning, the products of proteins hydrolysis were quantified by GelPro v3.1 software. The RAs of various IgG preparations were evaluated from a percentage decrease in the relative number of initial proteins compared to control (incubation of reaction mixtures in the absence of IgGs). All initial rates of proteins hydrolysis were estimated using the pseudo-first-order reaction conditions: linear regions of time dependencies in the case of concentrations of IgGs providing 20–40% of the hydrolysis of the proteins.

### 4.9. SDS-PAGE Analysis of Catalytic Activities

SDS-PAGE analysis of DNA-and protein-hydrolyzing activities of 2D2 mice IgGs was performed as in [34,35,36,37,38,39,40,41,42]. IgGs were pre-incubated at 25 °C for 30 min under non-reducing conditions (50 mM Tris-HCl buffer (pH 7.5), 1% SDS, and 10% glycerol). To restore the enzymatic activities of IgGs after SDS-PAGE, the gels were incubated for 1 h with 4 M urea at 20 °C and then washed 7 times (7–10 min) with H_2_O for removal of SDS. Then gel cross-sections of longitudinal slices (2–4 mm) were cut up and incubated with 50 μL of Tris-HCl buffer (50 mM, pH 7.5, containing 40 mM NaCl) for 5–7 days 4 °C, to allow IgGs refolding and elution from the gel. The gels were separated from solutions by centrifugation. The solutions obtained were used to analyze DNase and protease activities, as described above. Parallel longitudinal control gel stripes were used to find the position of IgGs on the gel by silver staining.

### 4.10. Statistical Analysis

The reported values are presented as the mean ± SD of at least three independent experiments for each mouse, averaged over two different groups of seven different male and female mice.

Some of the sets of samples did not match the Gaussian distribution. Therefore, the Mann-Whitney U test was utilized to estimate the differences between parameters analyzed; *p* < 0.05 was regarded statistically significant.

## 5. Conclusions

For the first time, we have analyzed the changes in the bone marrow stem cell differentiation profile, the levels of B, T, CD4, and CD8 lymphocytes in various organs, and the relative activities of IgG antibodies in the hydrolysis of DNA, MBP, MOG, and histones during slow spontaneous development of EAE in 2D2 mice. In addition, a comparison of changes in these parameters in 2D2 mice with those in two other strains of mice (Th and C57BL/6) predisposed to spontaneous and MOG-induced development of EAE was carried out for the first time. Some noticeable and very strong differences were found in the time variation of some parameters between male and female of 2D2 mice and male and female 2D2 line with those of Th and C57BL/6 mice.

A conclusion was made that a disturbance of the differentiation profile of bone marrow stem cells is common for all three EAE-prone lines of mice, leading to the production of autoantibodies-abzymes hydrolyzing DNA, myelin basic protein, MOG, and histones.

## Figures and Tables

**Figure 1 molecules-27-02195-f001:**
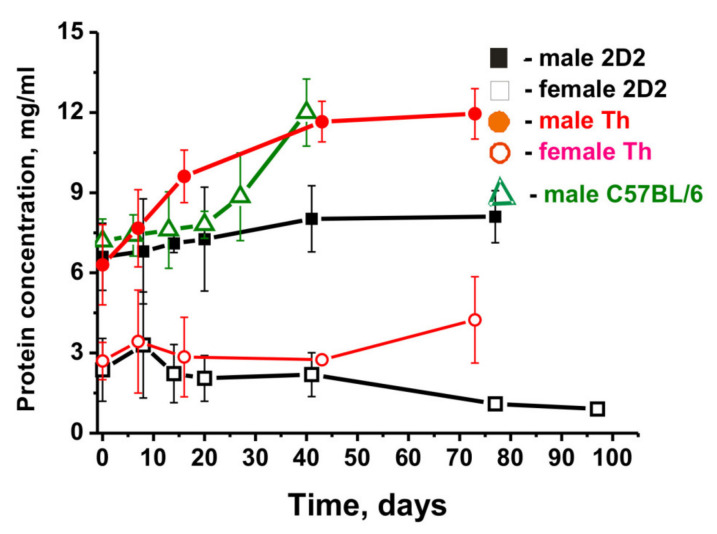
Relative changes in proteinuria during time-related development of EAE by 2D2 (black symbols), Th (red symbols) male and female, and C57BL/6 (green symbols) male mice. For comparison, the C57BL/6 [34,35,36,37] and Th [38,39] mice data were taken from our previously published articles. The duration of the experiment and the time of blood sampling for different strains of mice were different.

**Figure 2 molecules-27-02195-f002:**
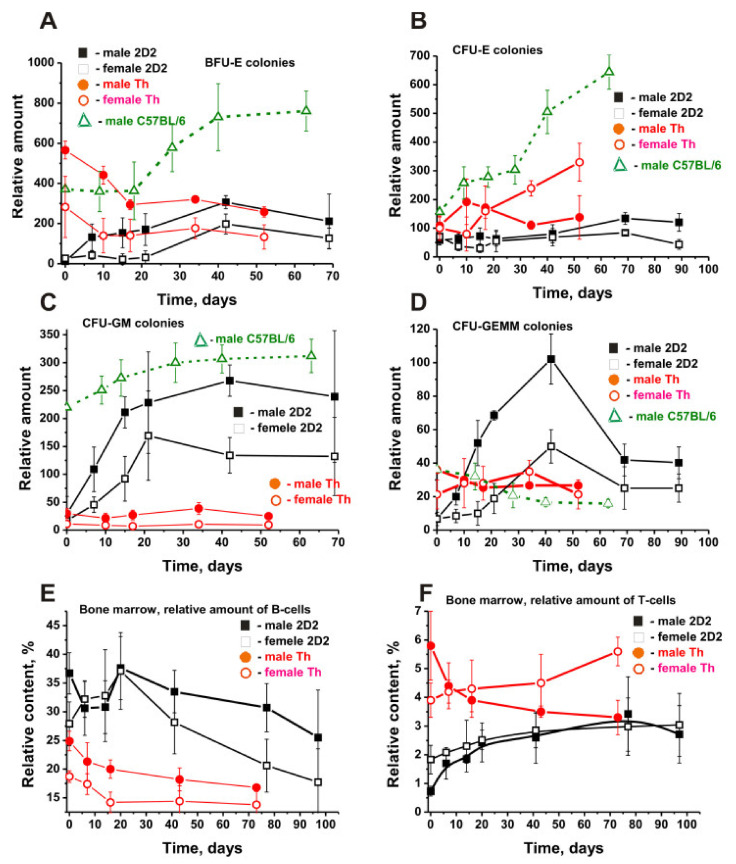
Changes during time-related development of spontaneous EAE in BFU-E (**A**), CFU-E (**B**), CFU-GM (**C**), and CFU-GEMM (**D**) cells as well as in BM relative amounts of B (**E**) and T lymphocytes (**F**) in 2D2 (black symbols), Th (red symbols), and C57BL/6 (green symbols) mice. The number of total cells is calculated for 15,000 bone marrow cells. All dependencies for all lines of males and females are marked in the Figure. For comparison, the C57BL/6 [34,35,36,37] and Th [38,39] mice data were taken from our previously published articles. The duration of the experiment and the time of blood sampling were different for different strains of mice.

**Figure 3 molecules-27-02195-f003:**
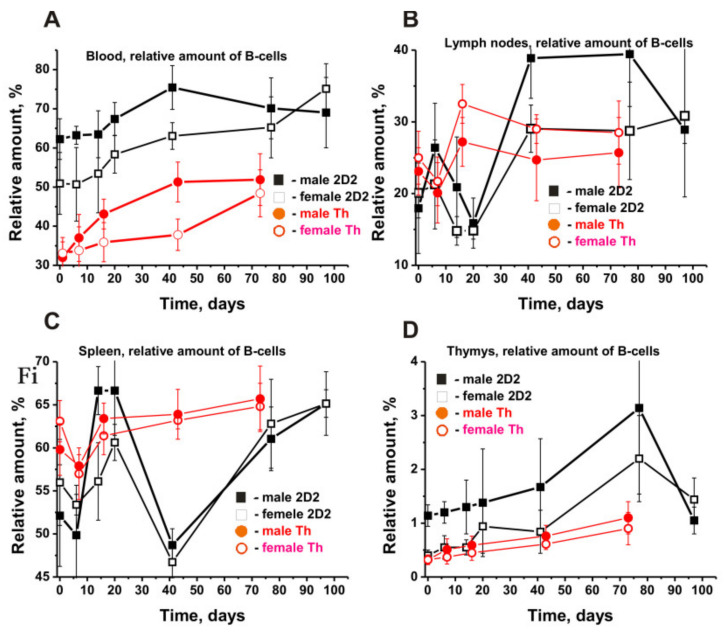
Time-related dependencies in the changes of relative numbers of B cells in different organs of 2D2 (black symbols) and Th (red symbols) males and females. All dependencies for all organs of various males and females are marked in the figures (**A**–**D**). For comparison, the data for Th mice were taken from our previously published articles [38,39]. The duration of the experiment and the time of blood sampling for different strains of mice were different.

**Figure 4 molecules-27-02195-f004:**
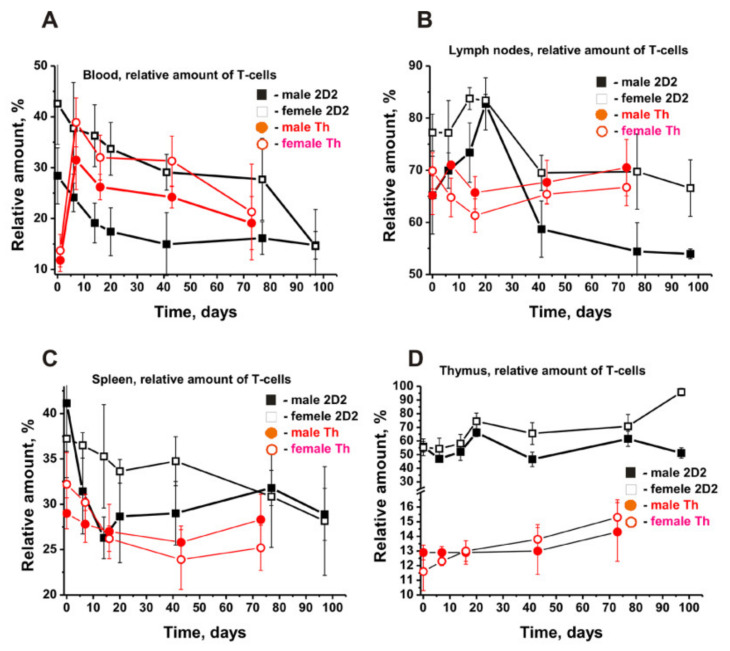
Time-related dependencies in the changes of relative numbers of T cells in various organs of male and female 2D2 (black symbols) and Th (red symbols) mice. All the dependencies for all organs of various males and females are given in (**A**–**D**) panels. For comparison, the data for Th mice were taken from our previously published articles [38,39]. The duration of the experiment and the time of blood sampling were different for different strains of mice.

**Figure 5 molecules-27-02195-f005:**
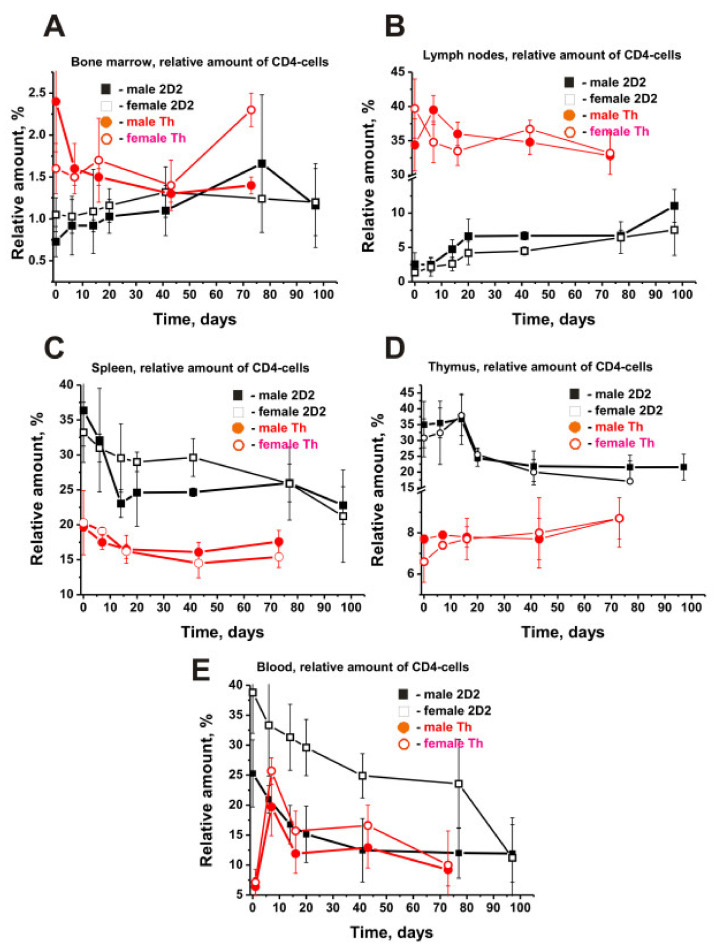
Time-related dependencies in the relative numbers of total CD4 cells in various organs of male and female 2D2 (black symbols) and Th (red symbols) mice. All dependencies for all organs of different males and females are marked in the Figure (**A**–**E**). For comparison, the data for Th mice were taken from our previously published articles [38,39]. The duration of the experiment and the time of blood sampling for different strains of mice were different.

**Figure 6 molecules-27-02195-f006:**
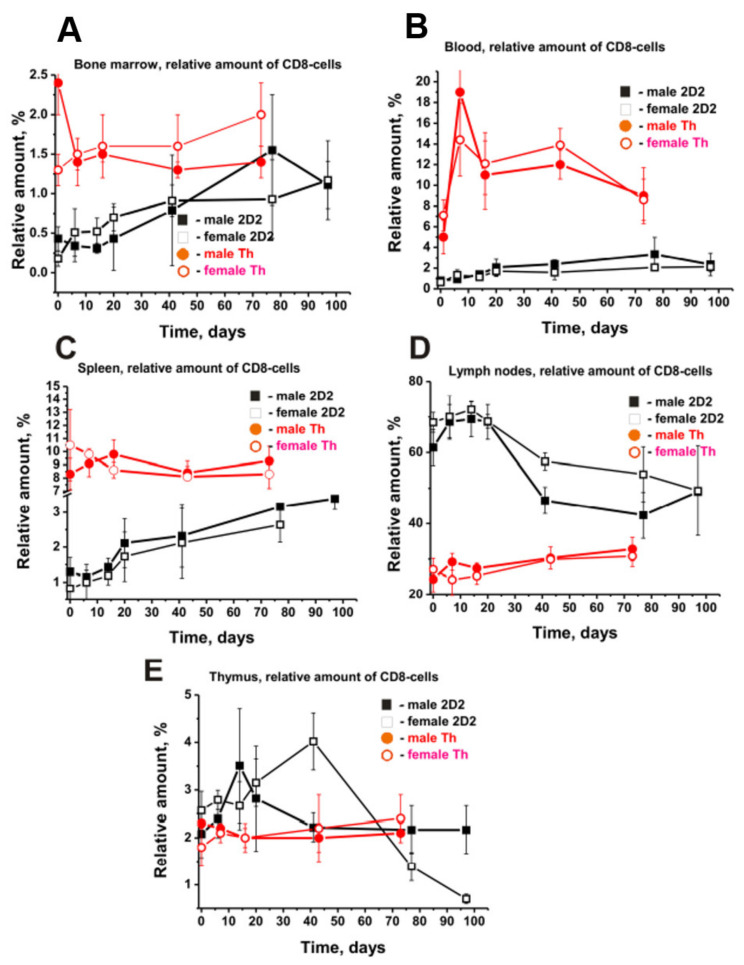
Time-related dependencies in the relative numbers of total CD8 cells in different organs of male and female 2D2 (black symbols) and Th (red symbols) mice. All dependencies for all organs of different males and females are marked in the Figure (**A**–**E**). For comparison, the data for Th mice were taken from our previously published articles [38,39]. The duration of the experiment and the time of blood sampling for different strains of mice were different.

**Figure 7 molecules-27-02195-f007:**
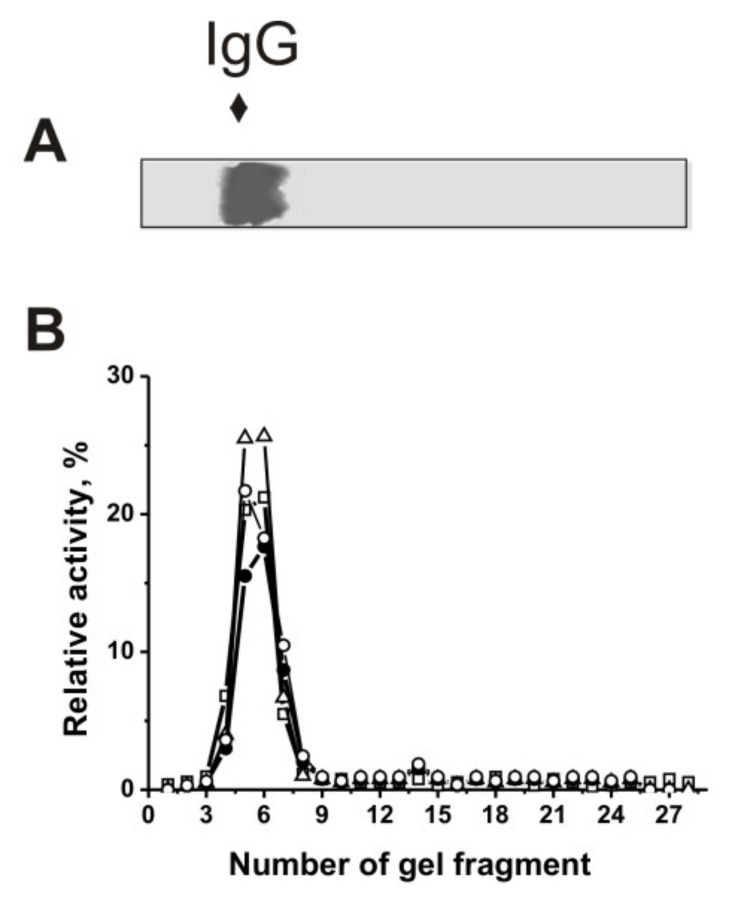
The IgG_mix_ (14 μg) homogeneity analysis by SDS-PAGE under non-reducing conditions in the absence of DTT (**A**); silver staining. Panel **A** demonstrates the position of IgGs. The relative activities (RA, %) in the hydrolysis of DNA (∆), MOG (□), MBP (●), and histones (o) were estimated using eluates of gel fragments (2–3 mm) (**B**). After substrates incubation for 24 h with eluates, complete hydrolysis of all substrates was taken for 100% (**B**). The errors of the relative activities estimation from two independent experiments did not exceed 7–10%.

**Figure 8 molecules-27-02195-f008:**
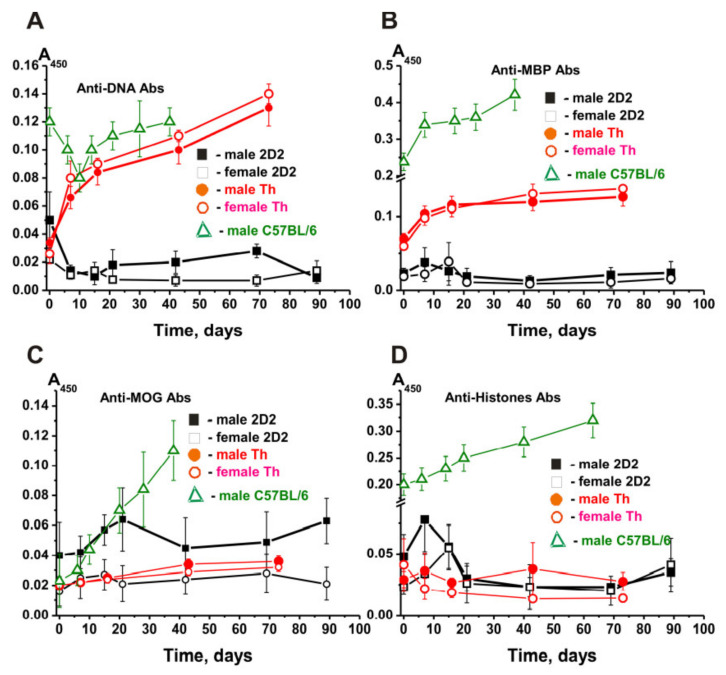
Time-related changes in the relative concentration of IgGs against DNA, MBP, MOG, and histones in purified Abs (**A**–**D**). Dependencies corresponding to 2D2 (black symbols), Th (red signs), and C57BL/6 (green symbols) males and females are shown in (**A**–**D**) panels. For comparison, the C57BL/6 [34,35,36,37] and Th [38,39] mice data were taken from our earlier published articles. The duration of the experiment and the time of blood sampling for different strains of mice were different.

**Figure 9 molecules-27-02195-f009:**
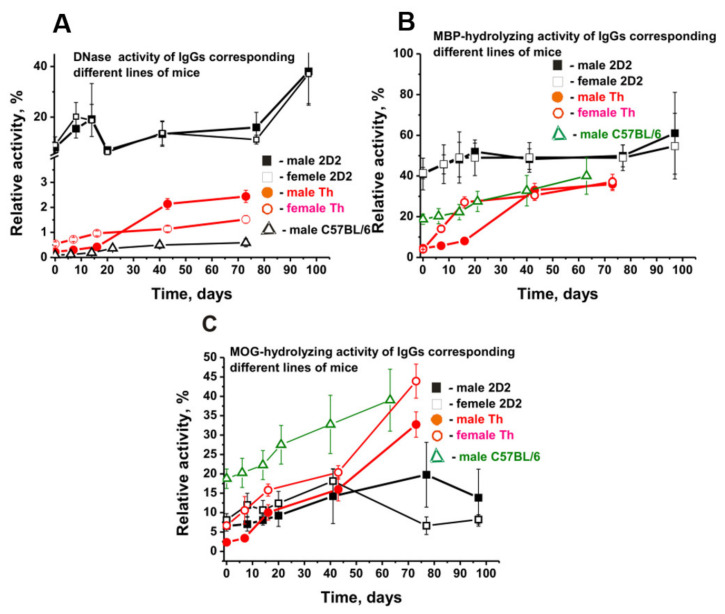
Time-related changes in the IgGs relative activities in the hydrolysis of DNA, MBP, and MOG (**A**–**C**). Dependencies corresponding to 2D2 (black symbols), Th (red signs), and C57BL/6 (green symbols) males and females are marked in the Figure. For comparison, the data for C57BL/6 [34,35,36,37] and Th [38,39] mice were taken from our earlier published articles. The duration of the experiment and the time of blood sampling for different strains of mice were different.

**Figure 10 molecules-27-02195-f010:**
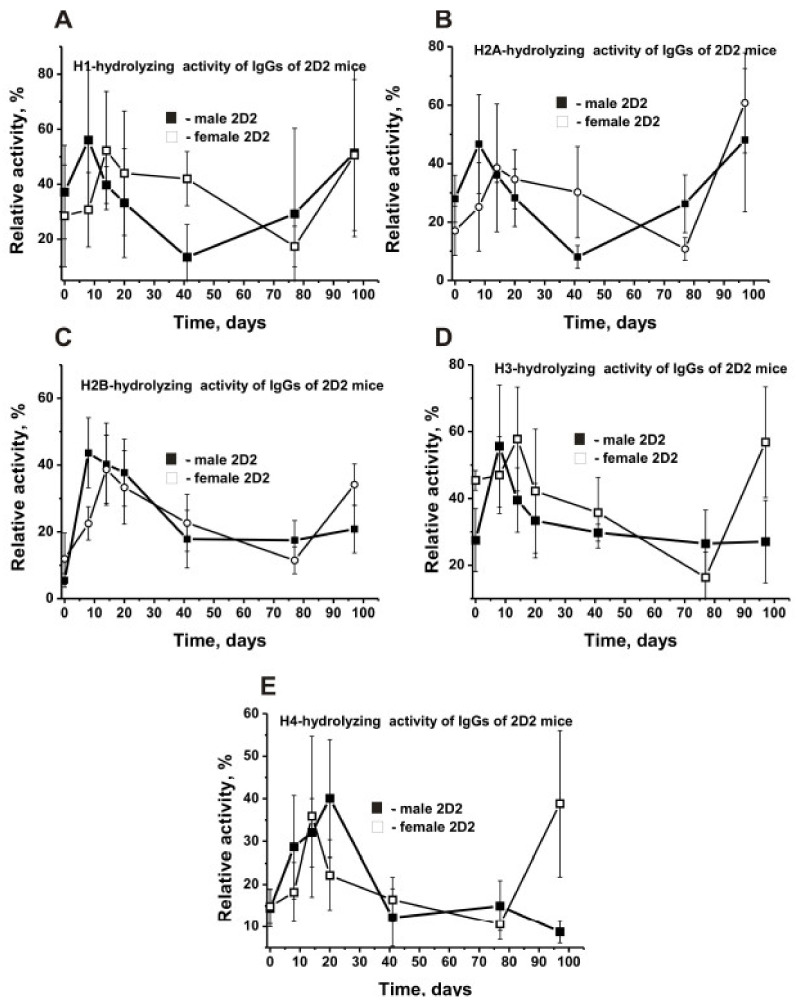
Time-related changes in the RA of IgGs in the hydrolysis of individual histones: H1 (**A**), H2A (**B**), H2B (**C**), H3 (**D**), and H4 (**E**). Dependencies corresponding to 2D2 male (■) and female (□) mice are marked in (**A**–**E**) panels.

**Table 1 molecules-27-02195-t001:** The average percentage content of different cells in various organs of male and female mice over a period of three months *.

Blood and Organs	The Relative Content of Different Cells in Various Organs of Mice, % **					Ratio of CD4 and CD8 Cells ***
	Sex	Total B Cells	Total T Cells	CD4 Cells	CD8 Cells	
Blood	male	62.3 ± 5.2	28.4 ± 5.2	25.3 ± 5.6	0.8 ± 0.3	31.6
	female	50.9 ± 7.9	42.6 ± 8.0	6.4 ± 1.5	0.6 ± 0.1	10.7
Ratio ^♣^		1.2	0.67	4.0	1.3	
Spleen	male	52.1. ± 2.9	41.1 ± 5.3	36.4 ± 5.1	1.3 ± 0.4	28.0
	female	56.0 ± 5.0	37.2 ± 4.3	33.2. ± 4.3	0.8 ± 0.3	41.5
Ratio ^♣^		0.93	1.1	1.1	1.6	
Bone marrow	male	36.7 ± 3.6	1.4 ± 0.17	0.73 ± 0.18	0.43 ± 0.12	1.7
	female	27.9 ± 3.8	1.8 ± 0.5	1.1 ± 0.27	0.18 ± 0.09	6.1
Ratio ^♣^		1.3	0.78	0.66	2.4	
Lymph nodes	male	18.0. ± 6.3	65.3 ± 7.5	2.5 ± 1.1	61.4 ± 5.0	0.04 (reverse 24.6)
	female	20.1 ± 3.5	77.5 ± 3.5	1.3 ± 0.3	68.5 ± 2.8	0.014 (reverse 52.7)
Ratio ^♣^		0.9	0.84	1.9	0.9	
Thymus	male	1.1 ± 0.2	55.8 ± 3.95	35.0 ± 5.3	2.1 ± 0.5	16.7
	female	0.32 ± 0.09	55.3 ± 3.1	30.8. ± 6.0	2.6 ± 0.4	11.8
Ratio ^♣^		3.4	1.01	1.1	0.8	

* Each group of male and female mice contained seven mice. ** For each mouse in each group, three independent measurements were performed, with the average (mean ± SD) values for seven mice given. *** The ratio of CD4/CD8 data is given. ^♣^ The ratio of data for male/female mice is given.

**Table 2 molecules-27-02195-t002:** The average relative protease activity of male and female mice IgGs at three months of age in the hydrolysis of different individual histones *.

The Relative Proteolytic Activity of IgGs, % **					
Sex	H1	H2A	H2B	H3	H4
male	37.1 ± 8.0	28.0 ± 8.0	5.5 ± 2.0	27.5 ± 9.3	14.4 ± 4.4
female	28.5 ± 9.0	16.0 ± 6.4	12.0 ± 6.0	45.4 ± 3.0	14.9 ± 4.1
Ratio	1.3	1.8	2.2	1.7	1.03

* Each group of male and female mice contained seven mice. ** For each mouse in each group, three independent measurements were performed, with the average (mean ± SD) for seven mice shown.

## Data Availability

The data supporting the findings of this study are available within the article.

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
