# Peer review of "Cell Differentiation and Proliferation in the Bone Marrow and Other Organs of 2D2 Mice during Spontaneous Development of EAE Leading to the Production of Abzymes"

_molecules, 2022, doi:10.3390/molecules27072195_

Round 1

Reviewer 1 Report

The authors addressed most of my comments. I recommend it for publication.

Reviewer 2 Report

I thank the authors for their response

This manuscript is a resubmission of an earlier submission. The following is a list of the peer review reports and author responses from that submission.

Round 1

Reviewer 1 Report

In this manuscript, Kseniya S. Aulova and colleagues evaluate different cellular and molecular markers in multiple sclerosis(MS)-prone mouse models including 2D2 TCR (TCRMOG), C57BL/6 and Th mice. The differentiation profiles of BFU-E, CFU-E, CFU-GM and CFU-GEMM colonies and T and B cells were longitudinally analyzed. Although this study could be interesting for most of researchers working with these mouse models, the conclusions are abstruse because the authors combine not only the immune cell analyses but also the variation of these cells according to the gender during the pathology progression. In conclusion, there is no clear... conclusion. 

Major concerns:
1.    English needs to be extensvely edited and the manuscript with red words is difficult to read
2.    Methods must be improved for the generation and analyses of BFU and CFU
3.    Methods must be improved for the analysis of the catalytic antibodies (abzymes) - How the authors are certain that the Ig enzymatic activity is not associated with contaminant co-purified with the IgG.
4.    Raw data obtained with Abzymes must be added in the manuscript (pictures of the gels for DNA degradation and western blots for protein degradation have to be shown in the main figures or supporting figures)
5.    The authors should add in the graph depicting the cell analyses, different clinical markers of the EAE progression, because cell analyses without clinical markers render difficult to appreciate the variation of the different analyzed markers and the pathology progression
6.    “Figure 7. The IgGmix homogeneity analysis by SDS-PAGE in the absence of DTT (A); silver staining. Panel A demonstrates Table 2. mm) (B). After their incubation for 24 h with eluates, complete cleavage of all substrates was taken for 100% (B). The errors of the relative activities estimation from two independent experiments did not exceed 7–10%. “ 
For instance, this figure legend is very difficult to understand. Could the authors improve their figure legends in general and be more accurate in the way they describe their figure?

Reviewer 2 Report

The manuscript by Aulova et al. describes a comparative analysis of 2D2 mice with C57BL/6 and Th mice in terms of T cells, B cells, and autoantibody production. They further state that in 2D2 mice, the bone marrow stem cells produce autoantibodies-abzymes which leads to the EAE development.

The manuscript has extensive grammatical errors and several spelling mistakes throughout which makes it hard to understand and follow the authors point of view. The introduction needs to be improved to provide a proper background and rationale for the study. The materials and methods are difficult to understand clearly. The results are very complex and unnecessarily lengthy with no proper statistics done or mentioned. The discussion and conclusion are also full of grammatical errors which makes it difficult to understand the authors hypothesis. Overall the manuscript is poorly written and explained